# Robotic versus Open Pyeloplasty: Perioperative and Functional Outcomes

**DOI:** 10.3390/jcm12072538

**Published:** 2023-03-28

**Authors:** Stefano Moretto, Carlo Gandi, Riccardo Bientinesi, Angelo Totaro, Filippo Marino, Filippo Gavi, Andrea Russo, Paola Aceto, Francesco Pierconti, Pierfrancesco Bassi, Emilio Sacco

**Affiliations:** 1Department of Urology, Università Cattolica del Sacro Cuore di Roma, Fondazione Policlinico Universitario Agostino Gemelli IRCCS, 00168 Rome, Italyemilio.sacco@policlinicogemelli.it (E.S.); 2Department of Emergency, Anesthesiological and Reanimation Sciences, Università Cattolica del Sacro Cuore di Roma, Fondazione Policlinico Universitario Agostino Gemelli IRCCS, 00168 Rome, Italy; 3Department of Woman and Child Health and Public Health, Unit of Pathology, Università Cattolica del Sacro Cuore di Roma, Fondazione Policlinico Universitario Agostino Gemelli IRCCS, 00168 Rome, Italy

**Keywords:** ureteropelvic junction obstruction, dismembered pyeloplasty, robot-assisted laparoscopic pyeloplasty, open pyeloplasty

## Abstract

We designed a retrospective study to assess the surgical and economic outcomes of robot-assisted laparoscopic pyeloplasty (RALP) compared with open pyeloplasty (OP), including consecutive patients suffering from ureteropelvic junction obstruction and operated on from January 2012 to January 2022 at a single center. Preoperative, intraoperative, and postoperative outcomes, including costs, were comparatively analyzed. The primary outcome was 3-month success, defined as symptom resolution and no obstruction upon diuretic renal scintigraphy. Overall, 91 patients were included (48 OP and 43 RALP). The success rate at 3 months was 93.0% and 83.3% in the RALP and OP group, respectively (*p* = 0.178), and the results remained stable at the last follow-up (35.4 ± 22.8 months and 56.0 ± 28.1 months, respectively). Intraoperative blood loss (*p* < 0.001), need for postoperative analgesics (*p* = 0.019) and antibiotics (*p* = 0.004), and early postoperative complication rate (*p* = 0.009) were significantly lower in the RALP group. None of the assessed variables were a predictor for failure. The mean total direct cost per surgical procedure and related hospital stay was 2373 € higher in the RALP group. RALP is an effective and safe treatment for ureteropelvic junction obstruction; however, further studies are needed to evaluate the cost-effectiveness of RALP, accounting for indirect costs and cost-saving with new surgical platforms.

## 1. Introduction

The ureteropelvic junction obstruction (UPJO) is characterized by functionally significant impairment of urinary transport at the ureteropelvic junction (UPJ) level caused by intrinsic or extrinsic obstruction. Most cases are congenital in origin (fibrosis and compression by aberrant crossing vessels); however, acquired conditions, such as kidney stones, may also result in UPJO [1]. UPJO can be diagnosed incidentally in otherwise asymptomatic patients or may present with acute and often recurrent renal colic or chronic low back pain. Other less specific manifestations are hematuria, urinary infections, pyelonephritis, and hypertension [2]. 

If left untreated, this condition can lead to hydronephrosis, which presents with dilatation of the renal pelvis and calyces. Hydronephrosis can lead to interstitial fibrosis, nephron leakage, and, ultimately, hydronephrotic atrophy. Therefore, early identification of signs and symptoms, accurate diagnosis, and timely treatment are of the utmost importance [3]. 

High-resolution diagnostic investigations, such as CT and MRI, provide anatomical details of the obstruction (including aberrant vessels) and can identify possible underlying causes, such as stones or urothelial tumors. However, sequential renal scintigraphy with diuretic testing is the most effective technique for diagnosing and quantifying obstruction and renal dysfunction; a radioisotope half-life of more than 20 min is consistent with obstruction [4].

Surgical management of UPJO has historically been performed with open dismembered pyeloplasty (OP), according to Anderson-Hynes, which was considered the standard gold treatment of UPJO, yielding a greater than 90% success rate [5]. 

With the introduction of endourologic techniques in the 1980s, less invasive techniques, such as antegrade and retrograde endopyelotomy, have been developed to minimize morbidity while attempting to achieve similar outcomes as open pyeloplasty [6].

Endoscopic placement of ureteric double-J stents should be considered as a first-line treatment in the management of primary hydronephrosis especially in children till 4 years of age, with success rate of 83.5% and without the need for conventional surgery [7].

In 1993, Schuessler et al. introduced the laparoscopic approach, which subsequently established itself worldwide as a valid alternative to OP, bringing the advantages of a minimally invasive technique in terms of a shorter hospital stay, lower morbidity, and less need for analgesics [8]. Robot-assisted laparoscopic pyeloplasty (RALP) was described in 1999 and has been recognized as an effective and reliable minimally invasive method for treating UPJO [9]. In 2002, Gettman et al. published the first series of nine patients undergoing RALP, reporting a success rate of 100% [10].

The robotic surgical platform is a valid surgical alternative in UPJO treatment, offering well-known advantages such as 3D visualization, magnification of the operative field, improved dexterity and ergonomics, and motion scaling with tremor reduction systems that facilitate dissection and intracorporeal suturing [11]. In addition, perioperative morbidity seems to be lower, and the learning curve is shorter than laparoscopic pyeloplasty [12]. The Da Vinci robotic system preserved the benefits of a minimally invasive technique in terms of reduced postoperative pain, reduced hospital stay time, and improved cosmetic outcomes, but the high cost of the robotic platform has led many to question whether the benefits of robotics technology are justified. 

Published data from comparative studies evaluating the robotic versus open approach remain scant. Therefore, we hereby report on an observational study comparing RALP versus OP regarding perioperative and functional outcomes and costs in a contemporary series of adult patients operated on at our urological referral center.

## 2. Materials and Methods

### 2.1. Study Design and Patient Recruitment

After local Institutional Ethic Committee approval (ID 5039; date of approval 25 July 2022), we performed an observational, retrospective, single-center study based on the electronic medical record review of consecutive RALP and OP cases using a prospectively maintained database. 

We defined UPJO by hydronephrosis on imaging (CT scan or MRI) and evidence of obstruction on diuretic renal scintigraphy (a radioisotope half-life of more than 20 min was consistent with obstruction).

Pyeloplasty was the first-line treatment in all surgically fit patients diagnosed with UPJO and accepting the procedure.

The recruitment period spanned from January 2012 to January 2022. 

The inclusion criteria were patient’s age ≥ 18 years, UPJO treated with dismembered pyeloplasty, and follow-up ≥ 6 months. 

The exclusion criteria were age < 18 years, therapeutic procedure other than dismembered pyeloplasty, redo procedure, patients with incomplete or missing records, or follow-up < 6 months to capture post-discharge complications. 

### 2.2. Data Collection

The following data were collected: Preoperative data: age, Charlson comorbidity index (CCI), body mass index (BMI), surgical history, and UPJO characteristics (side, signs and symptoms, preoperative and postoperative imaging, and prior nephrostomy or double-J stent).Intraoperative data: operating time, estimated bleeding, presence of crossing vessel or stones, complications, and conversion to open surgery.Postoperative data: length of hospital stay, analgesics and antibiotics requirement, early complications, late complications, and functional outcomes. All complications were classified with the Clavien–Dindo scale [13].

### 2.3. Patient Preparation and Positioning

To prevent venous thromboembolism, we combined the administration of pharmaco-thromboprophylaxis with low molecular weight heparin and graduated compression stockings. A double-J stent or, preferentially, a nephrostomy was placed preoperatively in patients with UPJO complicated by sepsis or acute renal failure. All patients were positioned in the lateral decubitus with the operative side facing up and rotated 45 degrees axially. The operating table was gently flexed at the waist to increase the exposure of the hip. We cared to anchor the patient to the operating table and tamp down all pressure points. We used 2 g Cefazolin as the standard one-shot antibiotic prophylaxis.

### 2.4. Surgical Procedures

All patients underwent a dismembered pyeloplasty. Whenever necessary, we performed stone surgery in the same session. We resected the UPJ if a narrow stricture was present. In the case of UPJO caused by obstructing crossing vessels, patients underwent an anterior transposition of the UPJ. We performed a kidney–psoas hitching if the anastomosis was not tension-free.

#### 2.4.1. Open Pyeloplasty

All patients underwent a lumbar incision at the 11th or 12th rib. Once we reached the retroperitoneum and incised the fascia of Gerota, we progressively isolated the kidney from the perirenal fat. After identifying and isolating the proximal ureter, we placed landmark stitches on the ureter before dissecting the renal pelvis just above the UPJ. Then, we spatulated the UPJ and the proximal ureter with a lateral incision. If we found markedly scarred tissue at the level of the UPJ; it was trimmed and sent for histopathological examination. Next, we performed the ureteropelvic anastomosis with an interrupted 4-0 Vicryl suture [3]. A double-J 6 Ch ureteral stent with the distal end in the bladder was inserted in an anterograde way using a guidewire after completion of the posterior side of the anastomosis. Finally, we placed a double laminar drain in situ for three days. Four expert surgeons performed the included procedures.

#### 2.4.2. Robot-Assisted Laparoscopic Pyeloplasty

We performed all of the procedures following a standardized four-arm, transperitoneal approach, plus 12 mm and 5 mm assistant ports (and an additional 5 mm port on the right side), using a 30° lens, fenestrated bipolar forceps, monopolar scissors, ProGrasp^TM^ forceps, and the da Vinci SI^®^ (until December 2015) or the da Vinci Xi^®^ robotic surgical platform (Intuitive Surgical, Inc., Sunnyvale, CA, USA). The 8 mm initial endoscopic trocar was placed laterally to the umbilicus, on the pararectal line, with the open Hasson technique; we used the pneumoperitoneum at a pressure of 12 mmHg, and we placed the remaining trocars as in Figure 1, 8 cm (about four fingerbreadths) apart from each other. This port configuration allows for, in our experience, optimal use of the robot, minimizing the external robotic arm clashing.

After docking the robot, we performed the retroperitoneum access and exposed the lower renal pole with the renal pelvis and the proximal ureter. The pelvis was incised just above the UPJ, and the ureter spatulated on its lateral side. Next, we performed pyeloplasty with two running Monocryl violet 4-0 sutures [14]. After completing the posterior suture, we inserted a double-J 6 Ch ureteral stent using a guidewire (Figure 2). We left a drain in situ for three days. Two expert robotic console surgeons performed the included procedures.

### 2.5. Postoperative Management and Follow Up

We mobilized patients on postoperative day one. We removed the bladder catheter on the third postoperative day.

We administered 1000 mg paracetamol upon request or to patients with a VAS score ≥4 so as to manage postoperative pain.

We removed the double-J stent four weeks after surgery by cystoscopy.

In the follow-up period, all patients had at least one diuretic renal scintigraphy three months after surgery to check for UPJO resolution; we then prescribed renal scintigraphy annually. The last follow-up was updated by a phone call by an independent assessor and using a data collection form.

### 2.6. Primary Outcome

The primary outcome was “success”, defined as a clinical response (symptoms resolution if present preoperatively) plus radiological response (no obstruction on follow-up diuretic renal scintigraphy). We defined failure as the persistence of symptoms and/or obstruction demonstrated on functional imaging and/or the requirement for a subsequent procedure for hydronephrosis (including nephrostomy and stenting). We classified failures as radiological only, radiological and clinical, or clinical only (symptoms persistence without demonstrated obstruction). We assessed surgical outcomes at 3 months and the last follow-up. In addition, we performed supplementary analysis by removing the first five cases of each surgeon involved in RALP surgery to account for a likely learning curve. 

### 2.7. Secondary Outcomes

Secondary outcomes were operative time, intraoperative estimated blood loss; conversion to open surgery rate; blood transfusion; administration of analgesics and antibiotics during the hospitalization; length of hospital stay; intraoperative, early, and late postoperative complications; recurrent symptoms rate; and nephrostomy/stent placement rate after surgery. 

In addition, we performed a retrospective perioperative cost analysis of the last three years using our institutional management software. We collected the following data related to the direct costs for each group of patients (OP and RALP):-the cost of diagnostic procedures performed during hospitalization.-the cost of the hospital stay.-the cost of the operating room.-the cost of devices used for surgery (including robotic equipment).

### 2.8. Statistical Analysis

We analyzed the demographic, clinical, and laboratory characteristics with descriptive statistic techniques. In addition, we used the Shapiro–Wilk test to investigate the normality in the distribution of the variables under exam.

We reported normally distributed continuous variables as mean ± standard deviation, otherwise as the median and interquartile range (IQR). We reported categorical variables as absolute and relative frequencies. We performed comparisons by chi-square test with Yates correction or Fisher’s exact test for categorical variables and by Student’s T-test or Mann–Whitney’s U test, as appropriate, for continuous variables. We deemed two-sided *p* < 0.05 to indicate statistical significance. We used no imputation techniques for the missing data. Instead, we undertook a univariate logistic regression analysis on all of the variables listed in Table 1 to find potential predictors of failure. 

We reported this study following the Strengthening the Reporting of Observational Studies in Epidemiology (STROBE) statement [15] and presented the STROBE checklist in the Appendix A. We conducted all analyses on statistical software R (R Cran, R core 2021).

## 3. Results

### 3.1. Patient Demographics 

Figure 3 shows the study flowchart. Ninety-one patients underwent dismembered pyeloplasty, of which 48 were OP (52.7%) and 43 were RALP (47.3%). The two cohorts had similar demographics, apart from CCI, which was higher in the OP group. We have summarized the characteristics of the patients in Table 1.

### 3.2. Primary Outcome 

RALP resulted in a numerically, but not statistically significant, higher 3-month success rate compared with OP: 40/43 (93.0%) vs. 40/48 (83.3%) (*p* = 0.178). Success rates remained stable at the last follow-up (35.4 ± 22.8 months for RALP and 56.0 ± 28.1 months for OP).

The robotic group had 3/43 (7.0%) failures, including two radiological only and one radiological and clinical failure. These patients underwent nephrectomy (1), retrograde endopyelotomy (1), and redo procedure (1). In the open group, surgery failure occurred in 8/48 (16.7%) cases, including five radiological-only, two radiological and clinical, and one clinical-only failure. These patients underwent redo procedures (5), nephrectomy (2), and retrograde endopyelotomy (1). 

### 3.3. Secondary Outcomes 

We have summarized the secondary outcomes in Table 2. 

RALP resulted in a statistically significant lower intraoperative bleeding, lower postoperative analgesics and antibiotics requirements, and lower early postoperative complications rate. However, the intraoperative complications rate, length of hospital stay, and late postoperative complications rate were only numerically lower in the RALP group. In addition, there were no conversions to open surgery during the robot-assisted procedures. 

In the supplementary analysis, removing the first five cases of each surgeon involved in RALP surgery, the results did not change (data not showed).

We summarized the cost data in Table 3. The budget gap between the robotic and open surgery was mainly related to the cost of the robotic equipment.

### 3.4. Predictors of Failures

Univariate logistic regression assessing potential variables predicting failure for the entire cohort revealed no statistically significant predictors (Table 4). However, we noted a trend toward better outcomes in patients with a lower BMI.

## 4. Discussion

The present study showed that RALP is safe and effective in treating UPJO. Furthermore, RALP resulted in a similar success rate to OP but was associated with advantages over OP in early postoperative complications. 

Our RALP success rate appears comparable with most of the other published series. However, the absence of a standardized definition of surgical success after pyeloplasty makes it challenging to compare different studies. Moreover, the clinical evaluation of patients is subjective, and many patients have unresolved flank pain associated with normal diuretic scintigraphy. In the literature, radiological success has been the preferred method for evaluating the effectiveness of pyeloplasty. However, the definition of radiologic success is not standardized and differs in many studies on RALP patients: Marien et al. considered the finding of T_1/2_ < 20 min at follow-up diuretic renal scintigraphy, reporting a 99% of radiological success rate [16], Mufarrij et al. considered a radiographic resolution of obstruction on first postoperative diuretic renal scan or excretory urogram, finding a 95.7% of radiological success rate [17]; and Masieri et al., instead, considered the resolution of hydronephrosis at follow-up ultrasonography, finding a 98% of radiological success rate [18]. 

Popelin et al. defined global success as clinical response plus radiographic evidence (at CT scan or furosemide-mercaptoacetyltriglycine-3 (MAG3) scintigraphy whenever CT was not fully diagnostic) of no further obstruction 3 months post-surgery. In their bi-center RALP experience, the radiological success rate was 96.2%, whereas the global success rate was 91% [19]. Autorino et al. examined the success rate between minimally invasive pyeloplasty and OP in the adult population, obtaining a success rate of 94% for both groups (OR: 0.96; 95% CI, 0.42–2.21; *p* = 0.93) [20].

Rasool et al. compared the mean operative times of OP, traditional laparoscopic pyeloplasty, and RALP, observing a statistically significant difference (*p* = 0.001) between traditional laparoscopic pyeloplasty (187.76 ± 22.1 min) and RALP (136.76 ± 25.1 min), but not between RALP and OP (132.06 ± 30.1 min) [21]. Consistently, in our case study, the mean operative time almost overlapped between OP and RALP (171.9 ± 54.8 min vs. 170.0 ± 49.5 min, respectively; *p* = 0.868). 

Intraoperative blood loss is expected to be reduced by minimally invasive approaches compared with the open techniques because of smaller surgical accesses and greater control of the operative field during the various surgical steps, reducing the risk of accidental organ or vessel injury. In the large multicenter study by Mufarrij et al., 117 patients in three centers underwent RALP, and the median blood loss reported was 57.4 mL (range 10–600) [17], in line with our results. The mean intraoperative blood loss was limited in both groups under comparison in our study, but was significantly higher (*p* = 0.001) in the OP group compared with the RALP group, in compliance with the results of other authors. Rasool et al. evaluated 102 patients who underwent pyeloplasty with open, laparoscopic, and robotic approaches, reporting a statistically significant difference in the mean blood loss between OP and RALP groups (86.47 ± 29.35 mL vs. 42.94 ± 20.77 mL, respectively; *p* < 0.001) [21]. 

The greater movement precision of the robotic arms in the various steps of pyeloplasty is expected to limit the intra-operative complication rate. Consistent with the literature, our study’s early complications rate (Clavien–Dindo grade ≥ 2) was statistically lower in the RALP group, while the late complications rate was only numerically lower [22,23,24]. Rasool et al. reported similar results, finding an approximately two-fold no statistically significant (*p* = 0.100) higher rate of postoperative complications in the OP group (11.7%) compared with those in the RALP group (5.88%) [21].

Postoperative pain is one of the most important factors to consider as it may affect the patient’s recovery time, length of hospital stay, and, consequently, hospital costs. With the introduction of minimally invasive techniques, due to the less invasive surgical approach and the presence of smaller surgical wounds, the incidence of pain in the immediate postoperative period and, consequently, the use of analgesics, have been significantly limited. Basatac et al. evaluated 56 patients undergoing RALP and OP, finding that analgesic use was significantly higher in the open group than in the robotic group (*p* = 0.02) [25]. Accordingly, in our study, the mean administered dosage of paracetamol during hospitalization significantly differed between the two groups (*p* = 0.019), being higher for the OP than for the RALP group.

Data on postoperative antibiotic use in patients undergoing pyeloplasty are scant in the literature. However, in the present study, a statistically significant difference (*p* = 0.004) in antibiotic requirement was found between the two groups, which occurred in 37.5% of open patients and only 9.3% of robotic patients. This difference in antibiotic requirement suggests a different postoperative course regarding complications and infectious episodes during hospital stay, confirmed by the higher incidence of the surgical site (12.5% vs. 0%) and wound infections (8.3% vs. 0%) in the OP group.

Autorino et al., in their meta-analysis comparing open and minimally invasive pyeloplasty techniques, obtained a difference in the weighted mean hospital stay of 2.68 days in favor of minimally invasive surgery [20]. However, consistent with a study by Sorensen et al. [26], in our study, the overall length of hospital stay showed no statistically significant difference (*p* = 0.131) between the two groups, which an underpowering of our study could explain. Despite the advantages that have emerged in the discussion of peri- and postoperative outcomes of RALP, the high costs of the robotic procedure remain a significant limitation to the large-scale application of this approach. Several studies comparing the costs of RALP with those of OP are available in the literature: Yu et al. found mean costs of $11,829 and $9520, respectively [27]; Link et al. also found that robotic treatment was more expensive than traditional laparoscopic approach by about 2.7 times due to the higher cost of robotic equipment [28].

In our study, the total costs were significantly higher for RALP than OP. This difference was mainly due to the cost of robotic disposables.

However, lower intraoperative and postoperative complication rates, less postoperative pain, and thus less use of drugs and diagnostic tests for the RALP group could compensate for the higher cost. These factors also imply a faster return to daily activities, positively impacting the resumption of work and social activities and, ultimately, lower indirect costs that we did not evaluate in our study. Finally, here are some tips and tricks to improve the results coming from our surgical experience. We suggest the following:Early preoperative removal (at least four weeks before) of a previously positioned stent, eventually replaced with a nephrostomy tube, to avoid inflamed tissues at the level of the UPJ.Limited isolation of the ureter and in situ anastomosis to avoid devascularization.Wide pelvis isolation to improve its mobilization.No coagulation on the ureteral and pelvis section margins.Landmark stitch on the ureter to avoid ureteral twisting.About 15 mm length of ureteral spatulation and pelvis opening, not including the scarred tissue that can be used for ureteral manipulation.Do not incorporate too much ureteral mucosa into the suture to avoid narrowing the anastomosis.Tension-free, watertight, and intubated anastomosis, usually performing a running suture, but an interrupted anastomosis, is a good alternative if there is too much tension.Kidney–psoas hitching if necessary to ensure a tension-free anastomosis, especially in the case of inflamed tissues (previous pyelonephritis, stone disease, or stenting).Avoid pelvis resection to shorten the suture length (pelvis tailoring is usually unnecessary).

The strengths of our study are the absence of major differences in demographics between the comparison groups, the standardized patient follow-up, and the multiple outcomes evaluation, adding more data to the scant literature in the field. Below are the main efforts to address potential sources of bias: Selection bias: we adopted extensive inclusion and exclusion criteria to include all patients undergoing pyeloplasty in our center.Observer bias: an independent researcher performed the assessment.Attrition bias: electronic medical record review of consecutive RALP and OP cases was performed based on a prospectively maintained database. We made numerous attempts to contact the patients if they could not be reached at the last follow-up, keeping the attrition bias to zero.Learning curve bias: we performed a supplementary analysis by removing the first five cases of each surgeon involved in RALP surgery, obtaining similar results.

The main limitations of this study are its observational, retrospective, and single-center nature; the relatively small sample size that may have made underpowered some comparisons; and the lack of validated self-administered questionnaires for the evaluation of postoperative outcomes. The small sample size did not allow for matching, subgrouping, or multivariable regression analyses. In addition, not all patients underwent renal scintigraphy in the same center. Furthermore, the different number of surgeons between the two groups and the different stages of the learning curve among the surgeons may have affected our study outcomes. 

Finally, the cost analysis did not include indirect costs, including all complications-related costs. While the “cost of hospital stay” and “cost of diagnostic procedures” captured the cost of complications occurring during the hospital stay, the costs of late complications and readmissions were challenging to collect from a referral center receiving patients from other Italian regions. According to our findings, the direction of this bias was in favor of the robotic approach.

## 5. Conclusions

RALP resulted in a similar success rate to OP, with a statistically significant lower intraoperative blood loss and early postoperative complication rate. Moreover, RALP entails improved cosmetic results, appearing to be a safe, effective, and minimally invasive treatment for UPJO. 

The budget gap between robotic and open surgery is mainly related to the cost of the robotic equipment. However, this cost gap will narrow with the introduction of last-generation and less expensive robotic platforms [29].

## Figures and Tables

**Figure 1 jcm-12-02538-f001:**
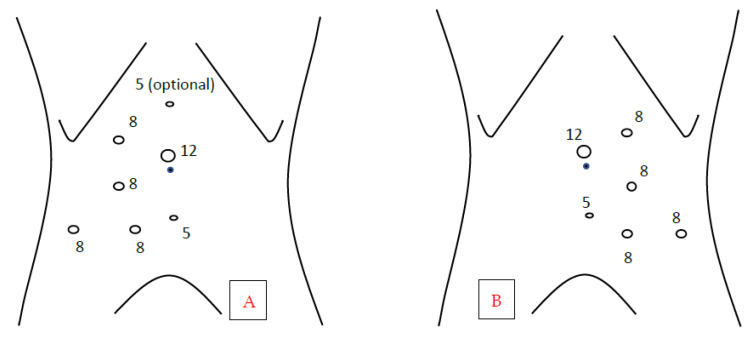
Ports placement template. (**A**): Trocar template for right RALP; (**B**): Trocar template for left RALP. 5 = 5 mm assistant port; 8 = 8 mm robotic port; 12 = 12 mm assistant port. Black dot: umbilicus.

**Figure 2 jcm-12-02538-f002:**
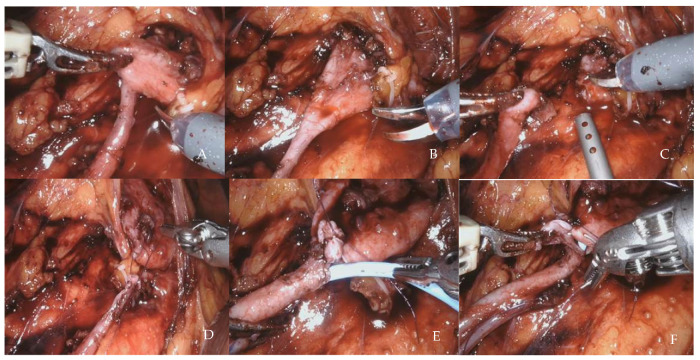
Robot-assisted laparoscopic pyeloplasty: main surgical steps. (**A**) Ureter traced proximally till pelvis. (**B**) Ureteropelvic junction delineated and tractioned. (**C**) Pelvis dismembered. (**D**) First stitch after lateral spatulation of the ureter. (**E**) Antegrade stenting after posterior suture was completed. (**F**) Anterior suture.

**Figure 3 jcm-12-02538-f003:**
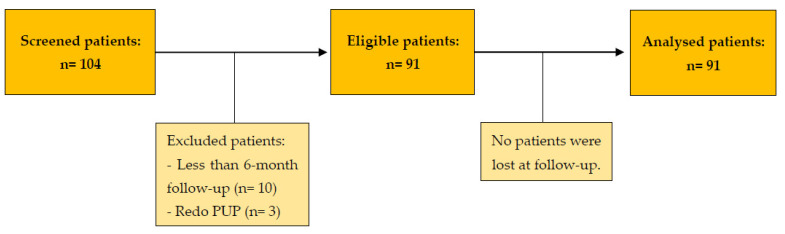
Study flow chart. n: number; PUP: pyelureteroplasty.

**Table 1 jcm-12-02538-t001:** Main demographic and preoperative data in the patients from the two groups. Values are means ± standard deviation or numbers with percentages.

	RALP Group (N= 43)	OP Group (N = 48)	*p* Value
**Gender**			
Male	16 (37.2)	25 (52.1)	
Female	27 (62.8)	23 (47.9)	0.225
**Age, years**	44 ± 19.3	49.7± 17,5	0.197
**BMI, kg/m^2^**	25.7 ± 2.9	25.5 ± 4.6	0.596
**CCI**	2.0 ± 2.1	1.6 ± 1.7	**0.015**
**Side**			
Right	25 (58.1)	30 (62.5)	
Left	18 (41.9)	18 (37.5)	0.883
**Presence of symptoms at diagnosis**	30 (69.8)	34 (70.8)	0.905
Renal colic	19 (44.1)	19 (39.5)	
Chronic Flank pain	10 (23.2)	13 (27.0)	
Recurrent UTIs	3 (7.0)	5 (10.4)	
Acute pyelonephritis	0 (0)	3 (8.8)	
Asymptomatic	13 (30.2)	14 (29. 1)	
**Preoperative double-j stent**	1 (2.3)	3 (6.3)	0.689
**Preoperative nephrostomy**	8 (18.6)	16 (33.3)	0.176
**Previous abdominal surgery**	10 (23.5)	19 (39.5)	0.148
**Concomitant stone disease**	7 (16.3)	16 (33.3)	0.103

RALP: robot-assisted laparoscopic pyeloplasty; OP: open pyeloplasty; N.: number; BMI: body mass index; CCI: Charlson comorbidity index; UTI: urinary tract infection. The bold used in the “*p*-value” column indicates a statistically significant *p*-value.

**Table 2 jcm-12-02538-t002:** Intraoperative and postoperative secondary outcomes. Values are means ± standard deviation or numbers with percentages.

	RALP Group	OP Group	*p* Value
	(n = 43)	(n = 48)	
**INTRAOPERATIVE OUTCOMES**			
**Operative time, min**	170.0 ± 49.5	171.9 ± 54.8	0.868
**Crossing vessels**	28 (65)	19 (39.6)	**0.026**
**Estimated blood loss, mL**	26.7 ± 11.4	80.2 ± 85.5	**<0.001**
**Blood transfusion**	0 (0)	0 (0)	
**Intraoperative complications**			
Renal vein lesion	0 (0)	1 (2.1)	0.341
**Conversion to open surgery**	0 (0)	n.a.	
**EARLY POSTOPERATIVE OUTCOMES**			
**Clavien–Dindo Classification** I II III IV	7 (16.2) 2 (4.6) 2 (4.6) 3 (7.0) 0 (0)	23 (47.9) 5 (10.4) 13 (27.1) 4 (8.3) 1 (2.1)	**0.002**
**Clavien–Dindo ≥ 2 complications**	5 (11.6)	18 (37.5)	**0.009**
Wound infections	0 (0)	4 (8.3)	
Surgical site infections	0 (0)	6 (12.5)	
Pneumonia	0 (0)	2 (4.2)	
Abdominal hematoma	1 (2.3)	0 (0)	
Hematuria	1 (2.3)	1 (2.1)	
Stent displacement	2 (4.6)	1 (2.1)	
Urinary leakage	1 (2.3)	2 (4.2)	
Urosepsis	0 (0)	2 (4.2)	
**Analgesic (paracetamol) requirement, mg**	2218.75 ± 2587.0	3531.9 ± 2842.6	**0.019**
**Patients requiring postoperative antibiotic course**	4 (9.3)	18 (37.5)	**0.004**
**Length of hospital stay, days**	7.7 ± 2.4	8.9 ± 4.5	0.131
**LATE POSTOPERATIVE OUTCOMES**			
**Complications, n. (%)**	3 (7.0)	9 (18.7)	0.178
Laparocele requiring surgical correction	0 (0)	1 (2.1)	0.922
Nephrostomy Placement, n. (%)	3 (7.0)	8 (16.7)	0.274

RALP: robot-assisted laparoscopic pyeloplasty; OP: open pyeloplasty; UPJ: ureteropelvic junction; n.a.: not available. We divided complications into early (≤30 days after pyeloplasty) or late (>30 days after pyeloplasty). The bold used in the “*p*-value” column indicates a statistically significant *p*-value.

**Table 3 jcm-12-02538-t003:** Cost data. Values are means ± standard deviation.

	RALP Group	OP Group	*p* Value
**Cost of diagnostic procedures performed during the hospitalization, €**	240.5 ± 146.5	445.5 ± 382.0	0.0009
**Cost of hospital stay, €**	2320.0 ± 630.4	3022.2 ± 1852.9	0.0153
**Cost of the operating room, €**	1854.3 ± 503.4	2460.4 ± 635.0	<0.0001
**Cost of devices used for surgery (including robotic equipment), €**	4286.0 ± 1083.7	398.9 ± 112.0	<0.0001
**Total cost, €**	8700.9 ± 1274.7	6327.1 ± 2404.4	<0.0001

RALP: robot-assisted laparoscopic pyeloplasty; OP: open pyeloplasty.

**Table 4 jcm-12-02538-t004:** Univariate logistic regression analysis evaluating factors associated with failure.

	OR	95% CI	*p* Value
**Sex (male vs. female)**	0.66	0.18–2.44	0.539
**Age (one-year increase)**	1.02	0.98–1.05	0.247
**BMI (unit increase)**	1.13	0.99–1.30	0.060
**CCI (unit increase)**	1.19	0.87–1.61	0.260
**Side (left versus right)**	2.91	0.81–11.92	0.109
**Presence of symptoms at diagnosis (yes vs. no)**	2.04	0.41–10.16	0.381
**Previous abdominal surgery (yes vs. no)**	1.26	0.35–4.47	0.718
**Preoperative double-j stent or nephrostomy (yes vs. no)**	1.14	0.22–5.92	0.871
**Crossing vessel (yes vs. no)**	0.75	0.21–2.67	0.661
**Concomitant stone disease (yes vs. no)**	2,87	0.78–10.50	0.111
**Postoperative infections (yes vs. no)**	1.71	0.45–6.47	0.426
**Surgical approach (robotic vs. open)**	0.39	0.09–1.59	0.192

BMI: body mass index; CCI: Charlson comorbidity index; UPJO: ureteropelvic junction obstruction; OR: odds ratio; CI: confidence interval.

## Data Availability

Data presented in this study are available on request from the corresponding authors.

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
