# Peer review of "Robotic versus Open Pyeloplasty: Perioperative and Functional Outcomes"

_jcm, 2023, doi:10.3390/jcm12072538_

Round 1

Reviewer 1 Report

In regards to the outcomes, I am not sure why the authors chose an odd 4 month period??
A lot of studies would look at complications at either a 1 month or 3/6 month period.

In addition, when comparing the costs of two procedures - the total treatment costs including costs of readmission or other costs related to complications from the procedures should have  to be added for a comprehensive assessments of costs; rather than just costs associated with surgery

Author Response

We thank the reviewers for their valuable and in-depth revision that allowed us to improve our paper. After that, we reported the corrections/changes we made in the revised version according to the reviewers' suggestions.

  • In regards to the outcomes, I am not sure why the authors chose an odd 4 month period?? A lot of studies would look at complications at either a 1 month or 3/6 month period.

Response: As per institutional policy, the first follow-up is scheduled three months after surgery. Due to the retrospective study design, patients had their first assessment between the third and fourth-month post-op. We do not see any serious bias in setting the first assessment at 3-mos which, as suggested, is a more commonly used endpoint.

  • When comparing the costs of two procedures - the total treatment costs including costs of readmission or other costs related to complications from the procedures should have to be added for a comprehensive assessment of costs; rather than just costs associated with surgery.

Response: We agree with the reviewer that a comprehensive cost analysis should also encompass complications-related costs; thus, we better highlighted the limitations of our analysis in the discussion section. In particular, we write: “Finally, the cost analysis did not include indirect costs, including all complications-related costs. While the “cost of hospital stay” and “cost of diagnostic procedures” captured the cost of complications occurring during the hospital stay, the costs of late complications and readmissions are challenging to be collected for a referral center receiving patients from other Italian regions. According to our findings, the direction of this bias is in favor of the robotic approach.”

Reviewer 2 Report

In their retrospective study, the authors compared the surgical and economic outcomes of robotic-assisted laparoscopic pyeloplasty (RALP) with open pyeloplasty (OP) in patients suffering from ureteropelvic junction obstruction. They concluded that RALP is an effective and safe treatment for ureteropelvic junction obstruction, but is significantly more expensive compared to OP.

I read the study with great interest. The study is well designed and written. I have several suggestions for improvement:

1. Introduction - The authors briefly presented the options for surgical treatment of ureteropelvic junction obstruction (UPJO). They did not mention endoscopic treatment with insertion of a double J prosthesis, which is especially important in neonates and pediatric patients. Please add a line about endoscopic treatment and the following reference: Pogorelić Z, Brković T, Budimir D, Todorić J, Košuljandić Đ, Jerončić A, Biočić M, Saraga M. Endoscopic placement of double-J ureteric stents in children as a treatment for primary hydronephrosis. Can J Urol. 2017;24(3):8853-8858.

2. The authors state that the study was approved by the local ethics committee (ID 5039). After the reference number (ID), they should add the date of approval, as well.

3. Authors have provided inclusion criteria, exclusion criteria should also be provided. Please revise.

4. Please indicate which statistical test was used to check normality of data distribution.

5. Authors should provide clear indications for surgery. They have defined UPJO, but they also need to give clear criteria for surgical treatment.

6. Paragraph ''2.4.1.. Open pyeloplasty'' - Please delete one period (2.4.1..)

7. The list of references should be updated. The majority of references are out of date. Please add some more recent studies.

8. The quality of the English language should be improved. The manuscript would benefit of editing by a native English speaker or a professional language editor to improve grammar and readability.

Reviewer 3 Report

The Authors present a comparison between Open pyeloplasty (OP) and Robot-assisted laparoscopic pyeloplasty (RALP) focusing on peri- and postoperative data. 

The paper is well written, clear and with a rigorous structure. English is good. Authors performed a proper statistical analysis. 

In my opinion the main limit to RALP's diffusion is the high costs of robotic equipments compared to laparoscopic pyeloplasty. This topic is well analysed in the paper.

Author Response

We thank the reviewers for their valuable and in-depth revision that allowed us to improve our paper. 

Round 2

Reviewer 2 Report

The authors adequately answered all questions raised by the peer review.